# CAMOU: Learning a Vehicle Camouflage for Physical Adversarial Attack on Object Detectors in the Wild

**Yang Zhang**[1], **Hassan Foroosh**[1], **Philip David**[2], and **Boqing Gong**[3]

[1] Department of Computer Science, University of Central Florida
[2] Computational and Information Sciences Directorate, U.S. Army Research Laboratory
[3] Tencent A.I. Lab
yangzhang@knights.ucf.edu, foroosh@cs.ucf.edu,
philip.j.david4.civ@mail.mil, boqinggo@outlook.com

## Abstract

In this paper, we conduct an intriguing experimental study about the physical adversarial attack on object detectors in the wild. In particular, we learn a camouflage pattern to hide vehicles from being detected by state-of-the-art convolutional neural network based detectors. Our approach alternates between two threads. In the first, we train a neural approximation function to imitate how a simulator applies a camouflage to vehicles and how a vehicle detector performs given images of the camouflaged vehicles. In the second, we minimize the approximated detection score by searching for the optimal camouflage. Experiments show that the learned camouflage can not only hide a vehicle from the image-based detectors under many test cases but also generalizes to different environments, vehicles, and object detectors.

## 1 Introduction

Is it possible to paint a unique pattern on a vehicle's body and hence hide it from being detected by surveillance cameras? We conjecture the answer is affirmative mainly for two reasons. First, deep neural networks will be widely used in modern surveillance and autonomous driving systems for automatic vehicle detection. Second, unfortunately, these neural networks are intriguingly vulnerable to adversarial examples (Akhtar & Mian, 2018).

Szegedy et al. (2013) found that adding imperceptible perturbations to clean images can result in the failure of neural networks trained for image classification. This motivates a rich line of work on developing defense techniques for the neural networks (Akhtar & Mian, 2018) and powerful attack methods to defeat those defenses (Athalye et al., 2018a). Moreover, the adversarial attack has been extended to other tasks, such as semantic segmentation (Arnab et al., 2018), object detection (Xie et al., 2017), image captioning (Chen et al., 2018a), etc.

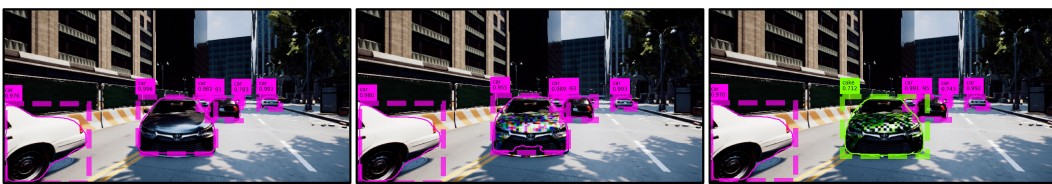

Figure 1: A Toyota Camry XLE in the center of the image fools the Mask R-CNN object detector after we apply the learned camouflage to it (on the right), whereas neither plain colors (on the left) nor a random camouflage (in the middle) is able to escape the Camry from being detected.

It is worth noting that the adversarial examples in the works mentioned above are not physical, i.e., the adversary directly manipulates image pixels. Although it is arguably more challenging to create physical adversarial objects than to produce adversarial images, some existing works have shown promising results with adversarial patches (Brown et al., 2017), stop signs (Eykholt et al., 2018b; Chen et al., 2018b), and small objects like baseballs and 3D turtle models (Athalye et al., 2018b).

To this end, we are reasonably optimistic about designing a special pattern to camouflage a 3D car, in order to make it difficult to detect by the deep learning based vehicle detectors.

It is undoubtedly challenging to run experiments in the real world considering financial and time constraints. In this paper, we instead demonstrate results using a simulation engine (Unreal) with a high-fidelity 3D sedan model and a 3D SUV. Fig. 1 shows that the vehicle in the simulation is photo-realistic such that, even covered with random camouflage, it can still be detected by the Mask R-CNN detector (He et al., 2017) trained on COCO (Lin et al., 2014).

The simulation engine enables us to test the physically adversarial cars under a considerable spectrum of environmental conditions: lighting, backgrounds, camera-to-object distances, viewing angles, occlusions, etc. In contrast, existing experiments on physical adversarial attacks are all executed in simplified scenarios. Eykholt et al. (2018b), Eykholt et al. (2018a), and Chen et al. (2018b) attack neural classifiers and detectors of stop signs. While projective transformations could be used to render the planar stop signs to various images, it is more involved to image the nonplanar 3D vehicles considered in this paper; we learn a neural approximation function instead. Athalye et al. (2018b) synthesize objects (e.g., baseball, turtle, etc.) which are adversarial within a small range of camera-to-object distances and viewing angles.

Given a 3D vehicle model in the simulation engine, we learn a camouflage for it by following the expectation over transformation (EoT) principle first formalized by Athalye et al. (2018b). The main idea is to consider a variety of transformations under which the camouflage can consistently hide the vehicle from a neural detector. A transformation imitates the imaging procedure and produces an image of the 3D vehicle model in the simulated environment. If a camouflage works under many transformations seen in the training phase, it is expected to also generalize to unseen transformations in the test phase.

One of the major challenges is that the simulator's image generation procedure is non-differentiable. A seemingly plausible solution is to train a neural network to approximate this procedure. The network takes as input the environment, a camouflage pattern, and the 3D vehicle model and outputs an image as close as possible to the one rendered by the simulator. Although this approach is viable, it is extremely difficult to generate high-resolution images. State-of-the-art methods (e.g., Render-Net (Nguyen-Phuoc et al., 2018)) can only generate simple 3D objects without any backgrounds.

We tackle the above challenge by drawing the following observation. In EoT (Athalye et al., 2018b; Chen et al., 2018b), the gradients propagate back to the physical object from the detector/classifier's decision values. If we jointly consider the object detector and the imaging procedure of the simulator as a whole black box, it is easier to learn a function to approximate this black box's behavior than to train the image generation neural network. Hence, we learn a substitute neural network which takes as input a camouflage, the vehicle model, and the environment and outputs the vehicle detector's decision value. Equipped with this substitute network, we can readily run the EoT algorithm (Athalye et al., 2018b) over our simulator in order to infer an adversarial camouflage for the vehicles.

Finally, we make some remarks about the significance and potential impact of this work. In the real world, multiclass visual object detection neural networks (He et al., 2017; Redmon & Farhadi, 2018) have become the cornerstone of multiple industrial applications, such as surveillance systems (Nair et al., 2018), autonomous driving (Huval et al., 2015), and military systems (Pellerin, 2017). Among these applications, cars are one of the most crucial objects. Attacking vehicle detectors in the physical world will be enormously valuable and impactful from the perspective of the malicious adversaries. Compared with the stop sign, it is legal in the United States to paint a car while defacing a stop sign is criminal. This poses a more significant threat to autonomous driving systems since anyone has access to perturb public machine learning based systems legally. This observation motivates us to focus our approach on cars. We limit our camouflage within the legally paintable car body parts, which means that we will leave discriminative visual cues, such as the tires, windows, grille, lights, etc. unaltered for the detectors.

## 2 RELATED WORK

### 2.1 ADVERSARIAL ATTACK AND ITS GENERALIZATION

Currently, the adversarial attack is powerful enough to attack image classification (Moosavi-Dezfooli et al., 2017), object detection (Arnab et al., 2018), semantic segmentation (Xie et al., 2017), audio recognition (Carlini & Wagner, 2018) and even bypass most of the defense mechanism (Athalye et al., 2018a). The mainstream adversarial machine learning research focuses on the in silico ones or the generalization within in silico (Liu et al., 2017). Such learned perturbations are practically unusable in the real world as shown in the experiments by Lu et al. (2017b), who found that almost all perturbation methods failed to prevent a detector from detecting real stop signs.

The first physical world adversarial attack by Kurakin et al. (2016) found perturbations remain effective on the printed paper. Eykholt et al. (2018b) found a way to train perturbations that remain effective against a classifier on real stop signs for different viewing angles and such. Athalye et al. (2018b) trained another perturbation that successfully attacks an image classifier on 3D-printed objects. However Lu et al. (2017c) found that Eykholt et al. (2018b)'s perturbation does not fool the object detectors YOLO9000 (Redmon & Farhadi, 2017) and Faster RCNN (Ren et al., 2015). They argue that fooling an image classifier is different and easier than fooling an object detector because the detector is able to propose object bounding boxes on its own. In the meanwhile, Lu et al. (2017a)'s and Chen et al. (2018b)'s work could be generalized to attack the stop sign detector in the physical world more effectively. However, all the above methods aim to perturb detecting the stop signs.

Blackbox attack is another relevant topic. Among the current blackbox attack literature, Papernot et al. (2017) trained a target model substitution based on the assumption that the gradient between the image and the perturbation is available. We do not have the gradient since the simulation is nondifferentiable. Chen et al. (2017) proposed a coordinate descent to attack the model. However, we found that our optimization problem is time-consuming, noisy (see Sec. G.2), empirically non-linear and non-convex (see Sec. H). Time constraints made this approach unavailable to us since it requires extensive evaluations during coordinate descent. Besides, coordinate descent generally requires precise evaluation at each data-point.

### 2.2 SIMULATION AIDED MACHINE LEARNING

Since the dawn of deep learning, data collection has always been of fundamental importance as deep learning model performance is generally correlated with the amount of training data used. Given the sometimes unrealistically expensive annotation costs, some machine learning researchers use synthetic data to train their models. This is especially true in computer vision applications since state-of-the-art computer-generated graphics are truly photo-realistic. Ros et al. (2016); Richter et al. (2017) proposed synthetic datasets for semantic segmentation. Gaidon et al. (2016) proposed virtual KITTI as a synthetic replica of the famous KITTI dataset (Geiger et al., 2012) for tracking, segmentation, etc. And Varol et al. (2017) proposed using synthetic data for human action learning. Zhang et al. (2017); Bousmalis et al. (2017) adapt RL-trained robot grasping models from the synthetic environment to the real environment. Tremblay et al. (2018) trained a detection network using the same Unreal engine that we are using.

## 3 PROBLEM STATEMENT

In this paper, we investigate *physical adversarial attack* on state-of-the-art neural network based object detectors. The objective is to find a camouflage pattern such that, when it is painted on the body of a vehicle, the Mask R-CNN (He et al., 2017) and YOLO (Redmon & Farhadi, 2018) detectors fail to detect the vehicle under a wide spectrum of variations (e.g., locations, lighting conditions, viewing angles, etc.). We learn the camouflage in a black-box fashion, without the need for accessing the detectors' network architecture or weights.

**Expectation-over-transformation (EoT).** We formalize the physical adversarial attack problem with the EoT framework (Athalye et al., 2018b). Denote by $t$ a transformation which converts a

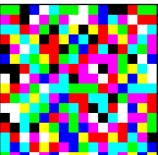 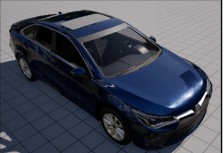 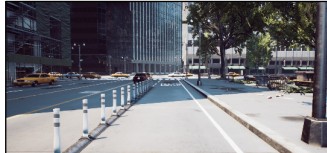 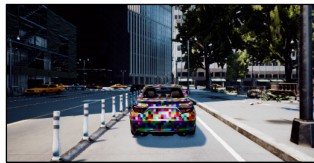

Figure 2: Example procedure of a transformation $t$. From left to right: a $16 \times 16$ camouflage, a high-fidelity vehicle, a location in our simulated environment, and an image due to this transformation over the camouflage.

camouflage pattern $c$ to a real photo. Such a transformation actually represents an involved procedure: paint the pattern to a vehicle's body, drive the vehicle to a location, configure a camera, and take a picture of the vehicle. The transformation conveniently abstracts away enormous factors (e.g., quality of painting, camera, etc.), each of which could play a role in this procedure. Denote by $V_t(c)$ the detection score (e.g., by Mask-RCNN) over an image which is a result of a transformation $t$ and a camouflage $c$. The physical adversarial attack problem is then described by

$$\arg \min_c \quad \mathbb{E}_{t \sim \mathcal{T}} \, V_t(c) \tag{1}$$

where $\mathcal{T}$ is a distribution over all the possible transformations $\{t\}$. In other words, we search for a camouflage $c$ that minimizes the vehicle detection score in expectation.

**Transformation in simulation.** Due to financial and time constraints, we conduct our study with the photo-realistic Unreal 4 game engine. It supplies us sufficient configuration parameters, such as the resolution and pattern of the camouflage, 3D models of vehicles, parameters of cameras and environments, etc. Fig. 2 shows a camouflage pattern, a high-fidelity 3D model of Toyota Camry, a corner of the virtual city used in this paper, and finally the picture taken by a camera after we apply the camouflage to the Camry and drive it to that corner — in other words, the rightmost image is the result of a certain transformation $t$ acting on the three images on the left of Fig. 2.

## 4 APPROACH

In this section, we present two key techniques for solving the problem $\min_c \; \mathbb{E}_t \, V_t(c)$ (cf. Eq. (1) and the text there). One is to estimate the expectation $\mathbb{E}_t$ by an empirical mean over many transformations. The other is to train a neural network to clone the joint behavior $V_t(c)$ of the black-box detector and the non-differentiable simulator.

### 4.1 SAMPLING TRANSFORMATIONS TO ESTIMATE $\mathbb{E}_t$

Recall that a transformation $t$ specifies a particular procedure from applying the camouflage $c$ to a vehicle until the corresponding image captured by a camera. In the context of the simulation engine, the camouflage $c$ is first programmed as textures, which are then warped onto the 3D model of an object. The simulator can teleport the object to different locations in the environment. The simulator also has multiple cameras to photograph the teleported object from various distances and viewing angles.

We identify some key factors involved in this procedure, including vehicle, location, camera-to-object distance, and viewing angle. The combinations of them also give rise to variations along other dimensions. For instance, the lighting condition changes from one location to another. The vehicle of interest is occluded to different degrees in the images captured by the cameras. Denote by $T_S$ all the sampled transformations that are used for learning the camouflage.

Fig. 3 illustrates the positions where the cameras are placed. We can see how the viewing angles and camera-to-object distances vary. The ones shown in the green color are used in the training. We left out some cameras shown in the red color for the testing. Since it is computationally expensive to utilize the cameras, we randomly arrange them in different heights and distances to create as much variations as possible instead of traversing all possible height-distance combinations.

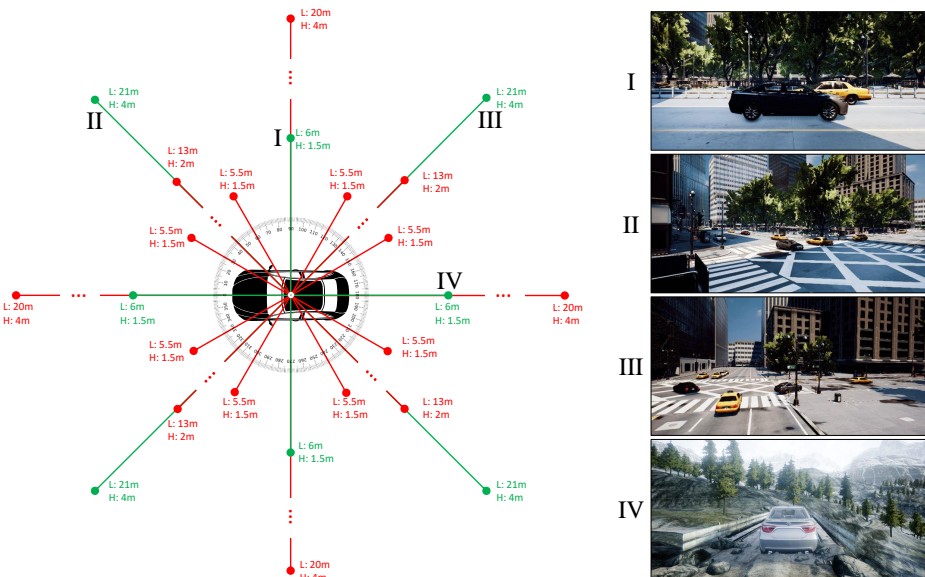

Figure 3: The camera setup as well as some exemplar locations. The cameras depicted in the green color are used to learn the camouflage while those in red are unseen cameras used for testing the camouflage's generalization. (H: camera height, L: vehicle-to-camera distance.)

## 4.2 LEARNING A CLONE NETWORK $V_\theta(c, t)$ TO APPROXIMATE $V_t(c)$

If we unroll the detection score $V_t(c)$, it contains two major components. The first renders an image based on the camouflage $c$ by following the procedure specified by the transformation $t$. The second obtains the detection score of a vehicle detector (e.g., Mask-RCNN) over the rendered image. The first component is non-differentiable while the second one could be a black box in practice. Therefore, we propose to consider them jointly as a single black box. Furthermore, we learn a clone neural network $V_\theta(c, t)$ to imitate the input-output behavior of this extended black box. As the transformation $t$ itself is involved and hard to represent, we instead input its consequence to the network: a background image and the cropped foreground of the vehicle.

Fig. 4b shows the network architecture. It takes as input a camouflage pattern $c$, the background image due to a sampled transformation $t$, and the cropped foreground. The network $V_\theta(c, t)$ has a single output to approximate the detection score $V_t(c)$.

To this end, we are ready to write down an approximate form of problem (1),

$$\arg\min_c \quad \frac{1}{|T_S|} \sum_{t \in T_S} V_\theta(c, t). \tag{2}$$

Thanks to the differentiable network $V_\theta(c, t)$ with respect to the camouflage $c$, we can solve the problem above by standard (stochastic) gradient descent.

It is important to note that the fidelity of problem (2) depends on the size and diversity of the sampled set of transformations $T_S$ as well as the quality of the clone network $V_\theta(c, t)$. It is straightforward to generate a large training set for the clone network by randomizing the camouflages and transformations and "labeling" the resulting images with the detection scores. However, if the optimal camouflage is unfortunately not covered by this training set, a discrepancy would occur when we solve problem (2). In other words, the network may fail to well approximate the detection scores at the region around the optimal camouflage. We address this discrepancy by an alternative learning algorithm as follows.

## 4.3 JOINTLY LEARNING THE CLONE NETWORK AND THE OPTIMAL CAMOUFLAGE

We alternatively learn the clone network $V_\theta(c, t)$ and solve the problem (2). Once a new camouflage pattern is found due to optimizing problem (2), it is converted to multiple images by the training

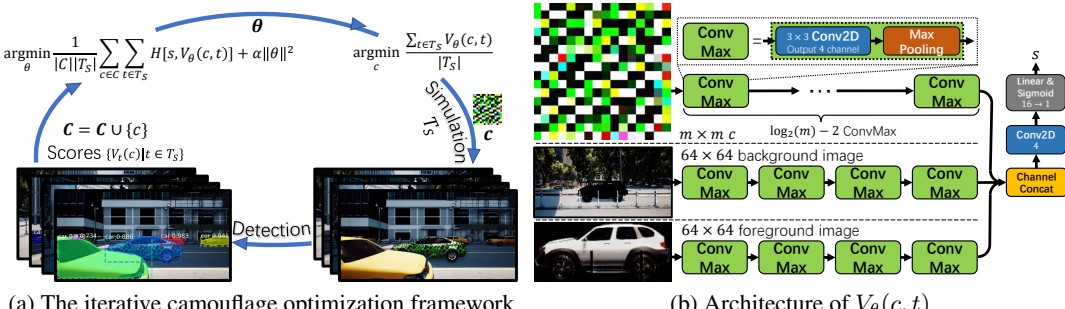

(a) The iterative camouflage optimization framework      (b) Architecture of $V_\theta(c, t)$

Figure 4: Overview of our optimization pipeline and the clone network $V_\theta(c, t)$.

transformations $T_S$. We obtain the detection score for each of the images by querying a detector. The camouflage pattern, along with the detection scores are then added to the training set for learning the clone network $V_\theta(c, t)$. Fig. 4a illustrates this process.

**Implementation details.** Denote by $H[p, q] := -p \log q - (1 - p) \log(1 - q)$ the cross-entropy loss. We alternately solve the following two problems,

$$\arg\min_\theta \ \frac{1}{|C||T_S|} \sum_{c \in C} \sum_{t \in T_S} H\big[s, V_\theta(c, t)\big] + \lambda \|\theta\|_2, \qquad \arg\min_c \ \frac{1}{|T|} \sum_{t \in T_S} H\big[0, V_\theta(c, t)\big] \quad (3)$$

where $C$ is the collection of all camouflages in the training set for learning the clone network $V_\theta(c, t)$, and $s := V_t(c)$ is the detection score corresponding to the camouflage $c$ and the transformation $t$. The $\ell_2$ regularization $\lambda \|\theta\|_2$ over the clone network's weights is essential. Without this term, the two loss functions may cause oscillations or degeneration of the solutions. We set $\lambda = 10$ in the experiments. Since the approximation accuracy of the clone network near the optimal camouflage is more important than in other regions, we weigh the newly added samples to the training set 10 times higher than the old ones at each iteration. Algorithm 1 in the Appendix gives more details.

## 5 EXPERIMENTS

Since the primary objective of this paper is to learn camouflage patterns that deceive vehicle detectors, we introduce two baseline camouflage patterns in the experiments: 6 most popular car colors and 800 random camouflages in different resolutions. We then analyze how the resolution of the camouflage affects the detection performance. For the vehicles, we employ two 3D models: a 2015 Toyota Camry and a virtual SUV. In addition to the main comparison results, we also test the transferability of the learned camouflage across different car models, environments, camera positions, and object detectors.

### 5.1 EXPERIMENT SETUP

We describe the detailed experimental setup in this section, including the simulator, vehicle detector, evaluation metrics, and the baseline camouflage patterns.

#### 5.1.1 THE SIMULATOR

As shown in Fig. 5a, we use the Unreal engine to build our first simulation environment from the photo-realistic DownTown environment. It is modeled after the downtown Manhattan in New York. There are skyscrapers, cars, traffic signs, a park, and roads in this environment, resembling a typical urban environment. We sample 32 different locations along the streets. Eight cameras perch at each location, each taking pictures of the size $720 \times 360$. The relative position of a camera is indexed by the viewing angle and camera-to-object distance, as shown in Fig. 3. In addition, we install another 16 cameras to test the generalization of the learned camouflage across viewing angles, distances, etc. In total, we use 18 locations for training and another 18 for testing. Note that the vehicles are invisible to some cameras due to occlusion.

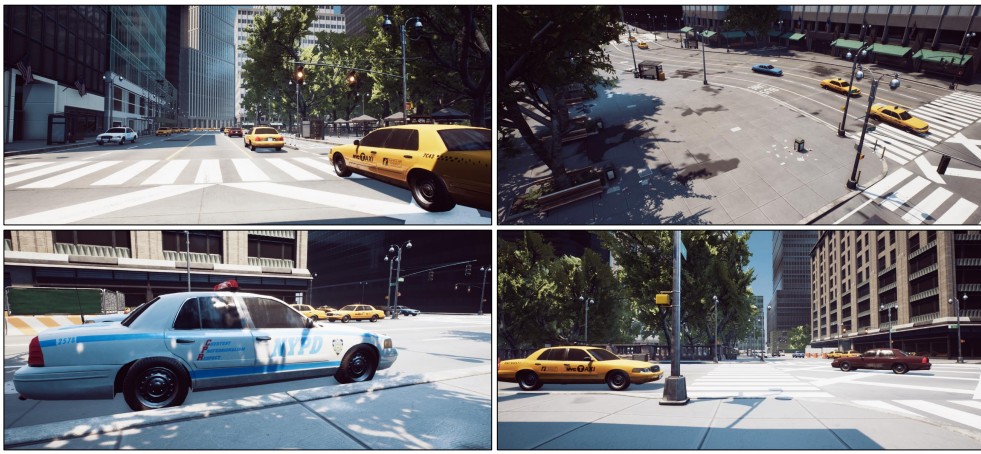

(a) An urban environment.

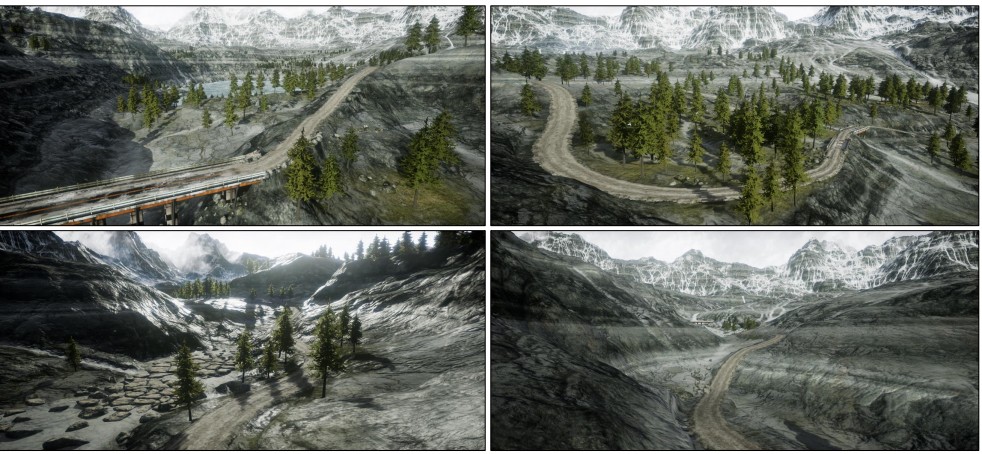

(b) A mountain environment.

Figure 5: The two environments we built to learn and test the vehicle camouflage. Zoom in for more details. These images are of the higher resolution but the same rendering quality (anti-aliasing level, shadow quality, rendering distance, texture resolution etc.) the detector perceived.

Our second environment is based on a totally different countryside scene called Landscape Mountains, shown in Fig. 5b. The roads lie on high-altitude mountains and cross bridges, forest, snow, and a lake. We use this scene to test the transferability of our camouflage across different environments; this scene is not used in the training. Like the DownTown environment, we sample 18 locations along the roads for the purpose of testing.

The two vehicles used in the experiments are shown in Fig. 6. One is a 2015 Toyota Camry XLE sold in the European Union. The other is a virtual SUV from AirSim (Shah et al., 2017). It is worth noting that the Toyota sedan appears multiple times in the MS-COCO dataset (Lin et al., 2014) (cf. some examples in Fig. 12). Since the object detectors are trained on MS-COCO, the Camry is more challenging to hide than the virtual SUV.

### 5.1.2 VEHICLE DETECTORS

We study two state-of-the-art detectors: Mask R-CNN (He et al., 2017) and YOLOv3-SPP (Redmon & Farhadi, 2018). Mask R-CNN is one of the most powerful publicly available object detectors; it currently ranks in the 4th place in the MS COCO detection leaderboard. Both detectors are pre-trained on MS COCO. For Mask R-CNN, we use the one implemented by Abdulla (2017). Its backbone network is ResNet-101 (He et al., 2016). YOLOv3 has comparable performance with

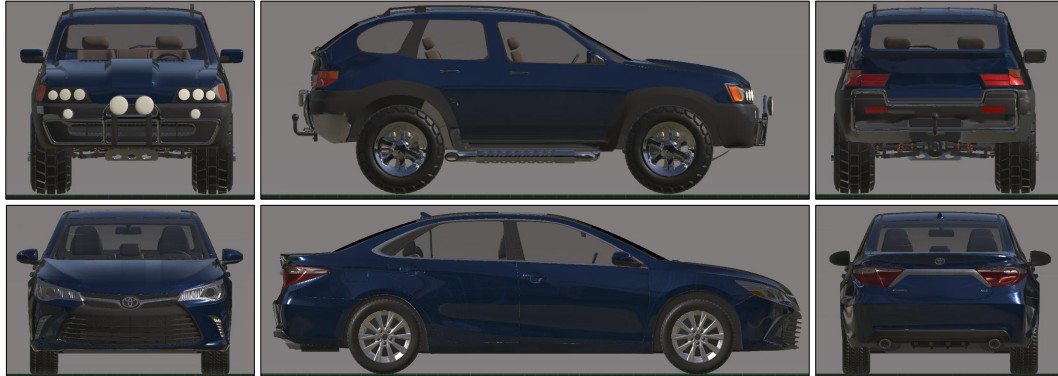

Figure 6: Orthogonal views of the Toyota Carmy XLE 2015 (second row) and the virtual SUV (first row) used in our simulation.

Mask R-CNN. Its network architecture is very different from Mask R-CNN's, causing challenges to the transfer of the camouflage between the two detectors. We use the spatial pyramid pooling (SPP) variant of YOLOv3 in the experiments.

In the rest of the paper, we experiment with Mask R-CNN except the transfer experiments in Sec. 5.4.

### 5.1.3    EVALUATION METRICS

We adopt two metrics to evaluate the detection performance. The first one is a variation of the Intersection over Union proposed by Everingham et al. (2015). The IoU between a predicted box and the groundtruth bounding box is defined as $IoU(A, B) = \frac{A \cap B}{A \cup B}$. Since IoU was originally proposed to evaluate the results of multi-object detection, as opposed to the single vehicle detection in our context, we modify it to the following to better capture the vehicle of interest: $\max_{p \in P} IoU(p, GT)$, where $P$ is the set of all detection proposals in an image. We average this quantity across all the test images and denote by mIoU the mean value.

Our second metric is precision at 0.5 or P@$0.5$. Everingham et al. (2015) set a 0.5 threshold for the detection IoU in the PASCAL VOC detection challenge to determine whether a detection is a hit or miss. We report the percentage of the hit detections out of all observations as our P@$0.5$. We also report the relative precision drop of the camouflage against the baseline colors whenever possible.

### 5.1.4    BASELINES

Our first baseline is when a vehicle is colored in popular real car colors. We select 6 basic car colors (red, black, silver, grey, blue, and white) which cover over 90% of the car colors worldwide (Axalta). We obtain their RGB values according to the X11 Color Names.

As the second baseline, we generate 800 random camouflages in different resolutions ranging in $\{2^i \times 2^i; i \in [1..8]\}$. Since we find that camouflages with strong contrasts work better, we generate half of the camouflages using RGB values $\in [0, 255]$ and the other half using RGB values $\in \{0, 255\}$. After we obtain these camouflages and the corresponding detection scores, we use those with proper resolutions to initialize the training set for the clone network $V_\theta(c, t)$.

The two baselines partially resolve the concern one might have that the learned camouflage successfully attacks the detector not because it exploits the CNN's structural weakness, but because it takes advantage of the domain gap between the real data used to train the detector and our simulated data used to test the detector. Results show that the baselines could not fail the detectors under most test cases, indicating that the detectors are fairly resilient to the domain gap between the simulated imagery and the real data (at least for the vehicle detection task).

### 5.2    RESOLUTION OF THE CAMOUFLAGES

We first report the random camouflages' performance on hiding the Camry from the Mast R-CNN detector in the DownTown environment. Fig. 7 shows the results. The first observation is that, al-

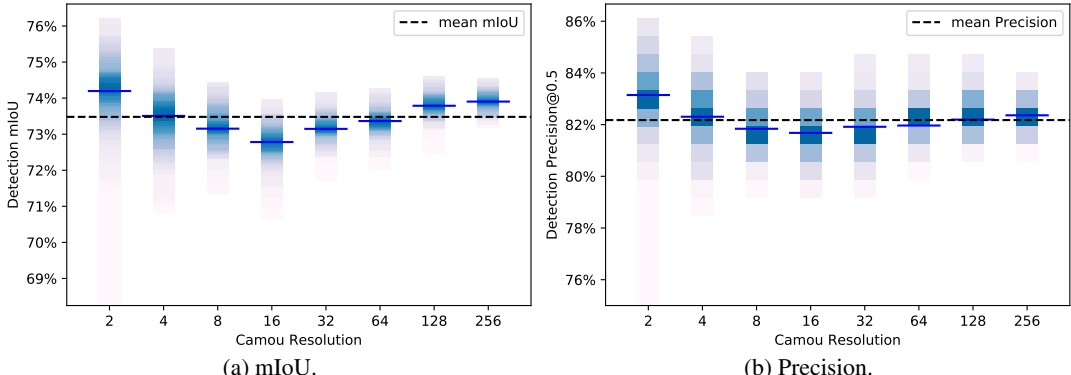

(a) mIoU.  (b) Precision.

Figure 7: The mIoU and P@0.5 of the 800 random camouflages in different resolutions on Camry in the DownTown environment. There are 100 camouflages per resolution.

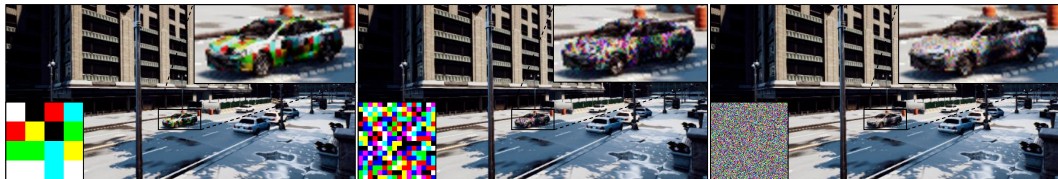

Figure 8: Visualization of three random camouflages of resolutions $4 \times 4$, $16 \times 16$, and $256 \times 256$, respectively, as well as the resulting images by the same camera.

though random camouflages are visually very different from conventional car paintings (cf. Fig. 8), the Mask R-CNN detector is still able to detect most of them. Another slightly counter-intuitive yet interesting observation is that the detector's performance does not always decrease as the camouflages' resolutions increase. This is probably because some fine-grained patterns displayed by the high-resolution camouflage become harder to observe from a distance. Since the high-resolution camouflage does not bring any additional benefits beyond the $16 \times 16$ resolution, we use camouflages of size $16 \times 16$ in the remaining experiments.

## 5.3 Camouflaging Toyota Camry in the urban environment

Here we report the results on detecting the Camry in the urban environment. Table 1 summarizes the results for the Camry respectively with the baseline colors, random camouflages, and our learned camouflage. We can see from the table that the Mask R-CNN is surprisingly robust against different types of random camouflages. The random camouflages' detection scores are close to the baseline colors'. Moreover, the standard deviation of the random camouflages' scores is also very low, indicating that the difference in the random camouflages does not really change the detector's performance. Note that we apply the camouflage to the body of the Camry, leaving tires, windows, grilles, etc. as informative visual cues to the detector. Despite that, our learned camouflage reduces the precision by around $30\%$ over baseline colors in both training and testing scenes in the urban environment.

| Camouflages | Training Scenes | | Testing Scenes | |
|---|---|---|---|---|
| | mIoU (%) | P@0.5 (%) | mIoU (%) | P@0.5 (%) |
| Baseline Colors | 76.14 | 84.40 | 72.88 | 77.57 |
| Random Camou | 73.48± 0.80 | 82.17± 1.20 | 67.79± 0.79 | 71.42± 1.20 |
| Ours | 57.69 | 62.14 | 53.64 | 52.17 |
| Relative Performance Drop | 24.23% | 26.37% | 26.39% | 32.74% |

Table 1: Mask R-CNN detection performance on the Camry in the urban environment.

| Camouflages | Training Scenes | | Testing Scenes | |
|---|---|---|---|---|
| | mIoU (%) | P@0.5 (%) | mIoU (%) | P@0.5 (%) |
| Baseline colors | 76.79 | 85.34 | 73.00 | 78.50 |
| Random Camou | 73.19±0.75 | 83.30±0.89 | 68.76±0.81 | 73.92±1.00 |
| Ours (YOLO trained) | 65.83 | 72.53 | 64.03 | 69.56 |
| Ours (Mask R-CNN trained) | 65.43 | 70.42 | 61.79 | 65.21 |
| Relative Performance Drop | 14.79% | 17.48% | 15.35% | 16.92% |

Table 2: YOLOv3-SPP detection performance on the Camry in the urban environment. In addition to the camouflage inferred for YOLO, we also include the YOLO detection results on the camouflage learned for Mask R-CNN.

**Virtual SUV.** Appendix A presents the results on the virtual SUV. The learned camouflage reduces Mask R-CNN's detection precision by around 40% from the precision for the baseline colors.

## 5.4 TRANSFERABILITY EXPERIMENTS

We show that the camouflage learned to attack Mask R-CNN can actually also defeat YOLOv3 to a certain degree. The results are reported in Table 2. Similarly, appendices A to D report the transferabilities of the learned camouflage across different environments, vehicles, camera viewing angles, and distances from the camera to the object.

## 5.5 QUALITATIVE RESULTS

We present some of our detection results on the baseline colors, random camouflages, and the learned camouflages for the Camry in Fig. 11.

We can draw a lot of interesting observations from the qualitative results which were hidden by the quantitative results. We find that there are 3 types of successful attacks: (1) Camouflages lower the objectiveness of the car and the car region is not proposed or only partially proposed as a candidate for the classifier of the detector; (2) The car region is successfully proposed but misclassified (e.g., to kite, cake, truck, or potted plant as shown in the examples) or the classification score is too low to pass the threshold for the detection score; (3) The car region is successfully proposed and classified, but the camouflage results in an incorrect detection which largely overlaps with the car (cf. the 5th row where the regions covering the car are detected as a car and a boat, respectively).

One can see that the context or background plays a vital role in object detection. Although some images capture the Camry by the same pose, the detector makes completely different predictions for them. Besides, these qualitative results imply that the detector works in a way different from human vision. In the Landscape environment, our learned camouflage has the strongest contrast to the background compared to other baseline patterns. However, the detector is still sometimes not able to detect the camouflaged car or SUV.

## 6 CONCLUSION

In this paper, we investigate whether it is possible to physically camouflage 3D objects of complex shapes, i.e., vehicles, in order to hide them from state-of-the-art object detectors. We conduct extensive experimental studies with a photo-realistic simulation engine. We propose to use a clone network to mimic the simulator and the detector's joint response to the 3D vehicles. Then, we infer a camouflage for a 3D vehicle by minimizing the output of the clone network. Our learned camouflage significantly reduces the detectability of a Toyota Camry and a SUV. Moreover, we find that the camouflage is transferable across different environments. For future work, We plan to look into possible ways to white-box the entire process so as to propose a more effective camouflage.

## ACKNOWLEDGMENT

This work was in part supported by the NSF grants IIS-1212948, IIS-1566511, and a gift from Uber Technologies Inc.

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

| Camouflages | Training Scenes | | Testing Scenes | |
|---|---|---|---|---|
| | mIoU (%) | P@0.5 (%) | mIoU (%) | P@0.5 (%) |
| Baseline Colors | 82.35 | 91.07 | 81.06 | 89.22 |
| Random Camou | 83.53±3.26 | 93.21±3.48 | 78.02±2.53 | 84.79±2.81 |
| Ours | 53.27 | 55.79 | 48.91 | 50.36 |
| Relative Performance Drop | 35.31% | 38.79% | 39.66% | 43.55% |

Table 3: Detection performance of camouflages on SUV in urban environment.

| Camouflages | Testing Scenes | |
|---|---|---|
| | mIoU (%) | P@0.5 (%) |
| Baseline Colors | 74.81 | 82.04 |
| Random Camou | 72.45±4.26 | 77.11±5.45 |
| Ours - Transferred | 40.39 | 43.26 |
| Relative Performance Drop | 46.00% | 47.26% |

Table 4: Detection performance of camouflages on Camry in Landscape environment. Note that this camouflage is pretrained in urban environment and then transferred without any finetuning.

## A    VIRTUAL SUV IN URBAN AREA

We then report the camouflage performance on the newly modeled SUV in the urban environment in Table 3.

Judging by the results, it is clear that the Mask R-CNN is a very generalized detector. The SUV's baseline color and random camouflage detection scores are even higher than the Camry correspondence although the detector has never seen it before. This might be because the SUV has much less polygon and is built to resemble the shape of a general SUV as shown in Fig. 6.

However, the price for not seeing this vehicle during training is that Mask R-CNN is more likely to be affected by the camouflage. The standard deviation of the random camouflage mIoU/ precision is higher (3.26/3.48 vs. 0.8/1.2) than Camry. And our camouflage achieves lower detection precision despite both baseline colors and random camouflage's detection scores are higher than Camry's. The detectability is reduced by almost 1.5 times of Camry's results in this case. This experiment shows that Mask R-CNN might well generalize to unseen vehicles, but it is easier to get attacked.

## B    TRANSFERABILITY ACROSS ENVIRONMENTS

Another important question is that what if we transfer the camouflage to a not only previously unseen but a totally different environment? To quantitatively answer this question, we build the Landscape environment (Fig. 5b) to test our camouflages trained in urban environment (Fig. 5a) using the Camry vehicle. The results are reported in Table. 4. Some qualitative results are shown in Fig. 11.

Given the barren landscape with few objects in sight, the detector detects the car better than it did in the urban environment for both baseline colors (82.04 vs. 77.57) and random camouflages (77.11 vs. 71.42) possibly due to the absence of distractions. However, our directly transferred camouflage is still able to beat both of them by more than 46% regarding both detection mIoU and precision without any fine-tuning.

It is interesting to notice that the Camry with grey color looks almost identical to the background in this environment (Fig. 11). However, it still could be perfectly detected. Meanwhile, our leaned camouflage results in far better stealth despite it has a sharp contrast to the background.

|  |  | Test | |
|---|---|---|---|
|  |  | SUV | Camry |
| Train | SUV | 50.36 | 47.44 |
|  | Camry | 58.39 | 52.17 |

Table 5: Camouflage transferability across vehicle reported in testing P@0.5 in urban environment.

| Camouflages | Testing Scenes | |
|---|---|---|
|  | mIoU (%) | P@0.5 (%) |
| Baseline Colors | 78.43 | 86.16 |
| Random Camou | 77.26±0.96 | 84.61±1.06 |
| Ours - Transferred | 67.74 | 73.14 |
| Relative Performance Drop | 13.62% | 15.11% |

Table 6: Detection performance of pretrained camouflages on Camry with urban environment in 16 unseen cameras.

## C  TRANSFERABILITY ACROSS VEHICLES

It will be impractical to retrain a specific camouflage for each vehicle whenever we need it. Hence it would be interesting to look into the transferability between vehicles in this scenario. We swap the camouflages of SUV and Camry and see how they would perform in the urban environment. We present the testing precision after swapping in Table. 5.

First, both camouflages are definitely transferable. It is also interesting to see that the Camry learned camouflage is not as good as the SUV learned camouflage even when being applied on the Camry itself. This might be due to the fact that the SUV resembles a car with more generic features and hence learning camouflage on SUV is less likely to encounter local minima during optimization.

## D  TRANSFERABILITY ACROSS VIEWING POSITION

One of the possibly most concerning questions is whether the learned camouflage is robust to the change of the camera position. To answer this question, we set up another 16 new cameras, as shown in Fig. 3, surrounding the vehicle in different relative locations. We then test the learned camouflage's performance on these new cameras. The results are shown in Table 6.

Our camouflage performance drop has a slightly decrease of 5% from Table 1. This indicates that the change of the camera locations would impact the performance, but the performance drop is still way beyond the standard deviation of random camouflages' scores. Given that the new camera views cover more perspectives as shown in Fig. 3, this result is reasonable.

## E  IMPACT OF CLONE NETWORK QUALITY

How does the clone network's quality affect the camouflage's performance? Is the alternative optimization necessary? We quantitatively show the first 300 simulation calls of our system in Fig. 9 and meanwhile evaluate the camouflages proposed by the clone network. Note that initially the clone network has already been trained with 800 random camouflages. However, the proposed camouflage's score does not fall until the new camouflages from iteration scheme join the optimization. This suggests that without our iterative optimization mechanism, the clone network could only find camouflage with mIoU around 70%, which is the same as the random camouflage. Those new samples serve as the hard samples. They gradually calibrate the clone network's global minima to $V_t()$'s and help it to generate better camouflages. Note that although the score descends quicker during the first 50 iterations, it does not find the global best camouflage until near the end (296th iteration). This graph also shows the classification score is a suitable choice to be minimized as it is highly correlated with mIoU.

Note that the system automatically re-initialize the clone network's parameters to prevent it from falling into local minima during the optimization whenever we add a new sample to the training set $C$. Hence, there are some spikes in the graph.

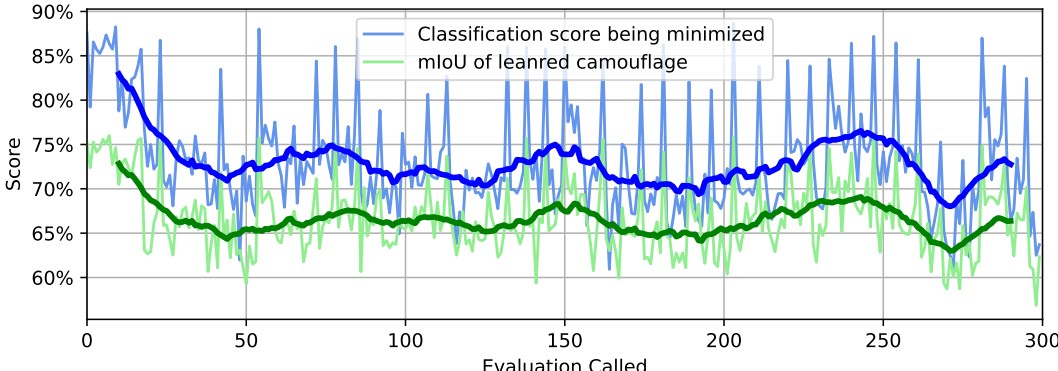

Figure 9: Clone network's learned camouflage's classification score and mIoU vs. Simulation called. We can see how the new samples helped the clone network to find the minimal.

## F    DETECTION ATTENTION

How exactly does our camouflage work against the detector? We partially answer this question by visualizing Mask-RCNN's completely/partially successful detection attention on our grey and camouflaged Camry. Since our Mask-RCNN implementation does not explicitly yield failed detection's prediction, we are unable to visualize the failed detection's attention w.r.t. the input image if the failed detection does not exist.

There are two main visualization approaches: Grad-CAM (Selvaraju et al., 2017) and saliency (Simonyan et al., 2013). Approximately speaking, grad-CAM visualizes the gradient of output w.r.t. penultimate (pre-fully-connected layer) convolutional layer feature map output; Saliency visualizes the gradient of output w.r.t. initial input image. Grad-CAM is generally considered superior as the last convolutional layer's feature map contains much more abstracted semantic information, leading to less noisy visualization. However, we find that it is hard to define the single "penultimate layer" in Mask-RCNN: It has multiple penultimate layers, tracing back to different stages of the network, prior to the ROI pooling layer. Each of those penultimate layers contains varying levels of information. We choose to use saliency in this case.

It is clear how to define the "attention" in the image classification scenario: It is the gradient heatmap of the classification score scalar w.r.t. the entire input image. On the other hand, an end-to-end detection neural network often yields multiple structure predictions. Each structure contains bounding box and classification score, etc. We choose to visualize the gradient of the best bounding-box's classification score w.r.t. the input image.

Our visualizations are presented in Fig. 10. It is surprising that the window, roof and upper bodies are playing the predominant role in car detection. Upper body attention exists even when the upper body is not included in the detection bounding box (row 3). Given that the classification stage makes the classification decision based on the proposed feature map box region, such attention must have been already included in the detection proposal before the ROI layer, i.e. detection stage. This may also explain why it is easier to fail the front-viewing and the rear-view detectors: front-view (row 1) and the rear-view (row 5) detectors place their attention on the hood and trunk, where camouflage pattern is presented. A partially successful attack (row 4) was carried out by wiping out detector's attention on the car hood. On the other hand, the side-view detector is harder to attack since the detector merely places any attention of the car body (row 2) where the camouflages are mainly located. However, our camouflages on the roof could still fail the side-view detector partially (row 3).

Since we only visualize the (partially) successful detections, there are still many cases to explore.

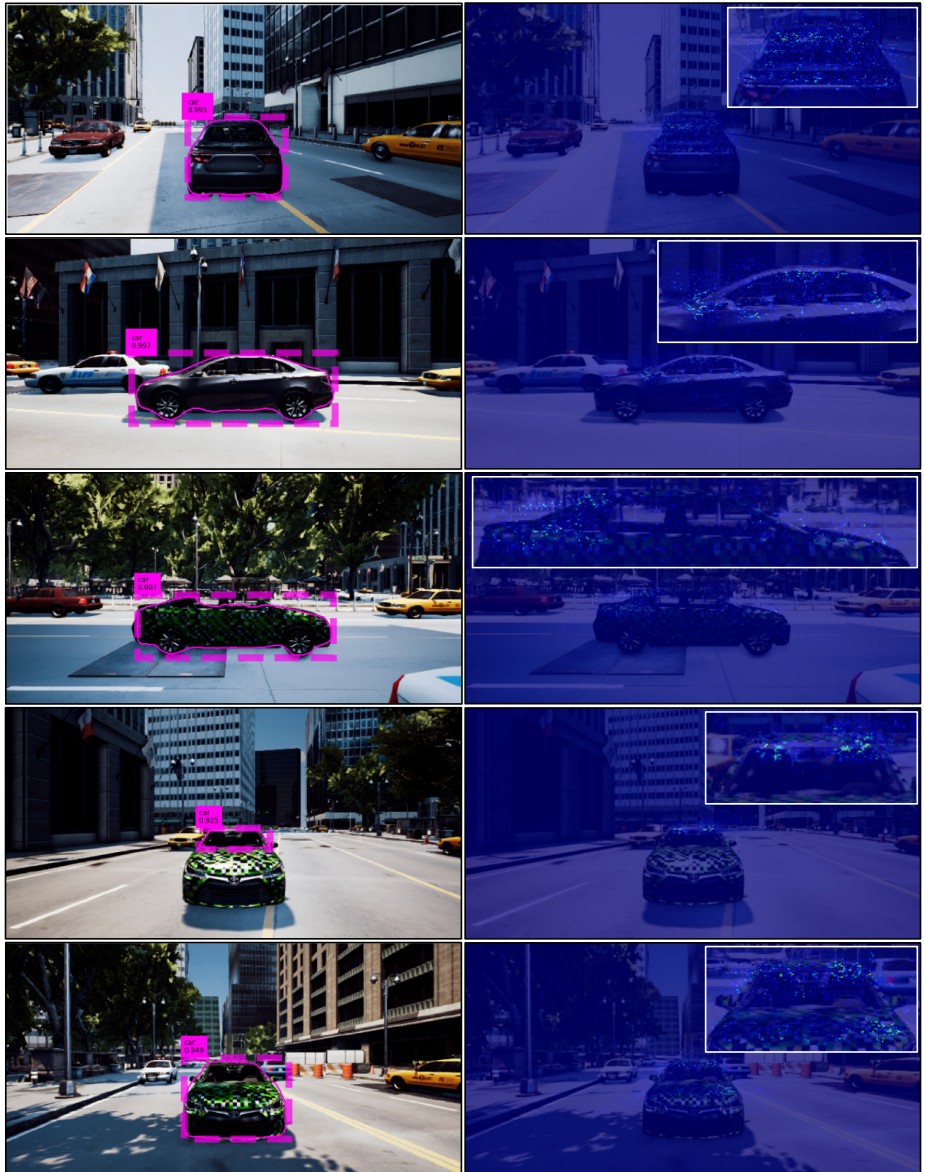

Figure 10: The best detections in each image and the gradient heatmap of their classification scores w.r.t. the input images. The detector places its attention predominantly on the upper car body, i.e., roof, hood, trunk, and windows.

## G  SIMULATION SETUP

### G.1  SIMULATION IMPLEMENTATION

The iterative optimization framework works on two Nvidia GTX 1080 Ti in our experiment. We use one to run the detector and another one to run the simulation and training/ prediction of evaluation network. The simulation was implemented partially using AirSim by Shah et al. (2017) and UnrealEnginePython. All submodules are implemented asynchronously to run in parallel and are communicating with each other using RPC/ RPyC. All the camouflages and the baseline colors are implemented either using or based on the official Unreal Automotive Material. Each evaluation in the simulation, which is the most time-consuming part, takes around 15 to 20 second.

## G.2 SIMULATION ERROR

Despite our best effort, we observe $V_t(c)$ come with a standard deviation of $0.008$ due to the inherent and mostly necessary random processes in the rendering (i.e., Monte Carlo in path tracing, etc.). This unfortunately makes $V_t(c)$ a noisy function. We reduce this error by repeat sampling $V_t(c)$ 5 times whenever we use it.

## H NON-LINEARITY AND NON-CONVEXITY

Since this is a blackbox optimization problem, it is important to examine some important features of the $\mathbb{E}_t V_t(\cdot)$. We first verify its convexity via the convexity definition. We test the convexity of $\mathbb{E}_t V_t(\cdot)$ by testing the convexity of subsampled correspondence $\frac{1}{|T_S|} \sum_{t \in T_S} V_t(\cdot)$ via:

$$\forall c_1, c_2 \in C: \quad \frac{1}{|T_S|} \sum_{t \in T_S} V_t(\frac{c_1 + c_2}{2}) \leq \frac{1}{|T_S|} \sum_{t \in T_S} V_t \frac{V_{T_S}(c_1) + V_{T_S}(c_2)}{2} \tag{4}$$

where $C$ is a set of camouflages. We sampled 1000 pairs of random camouflages from $C$ and half of them do not meet the equation. Hence $\frac{1}{|T_S|} \sum_{t \in T_S} V_t(\cdot)$ is nonconvex.

Besides, we find a simple linear MLP is insufficient to approximate $\frac{1}{|T_S|} \sum_{t \in T_S} V_t(\cdot)$, which empirically shows it is nonlinear.

## I SUPPLEMENTAL FIGURES

---

**Algorithm 1:** Iterative Object Camouflage Learning

---

**Input** : Clone network parameter $V_\theta(\cdot)$; Simulation and detection $V_{T_S}(\cdot)$; Transformation $T_S$ which are parameterized as rendered background and foreground images; Regularization tradeoff $\alpha$; Random camouflage set $C_R$.

1 Initialize $V_\theta$ with random weights $\theta$
2 Set score record $s^* \leftarrow +\infty$
3 $C \leftarrow C_R$
4 **repeat**
5      $\theta \leftarrow \arg\min_\theta \frac{1}{|C||T_S|} \sum_{c \in C} \sum_{t \in T_S} H[s, V_\theta(c, t)] + \lambda\|\theta\|_2$
6      $c' \leftarrow \arg\min_c \frac{1}{|T|} \sum_{t \in T_S} H[0, V_\theta(c, t)]$
7      $s' \leftarrow \{V_t(c) | t \in T\}$
8      $C \leftarrow C \cup \{c'\}$
9      **if** mean$(s') < s^*$ **then**
10          $s^* \leftarrow$ mean$(s')$
11          $c^* \leftarrow c'$
12 **until** Reach maximum training steps
**output:** Best learned camouflage $c^*$

---

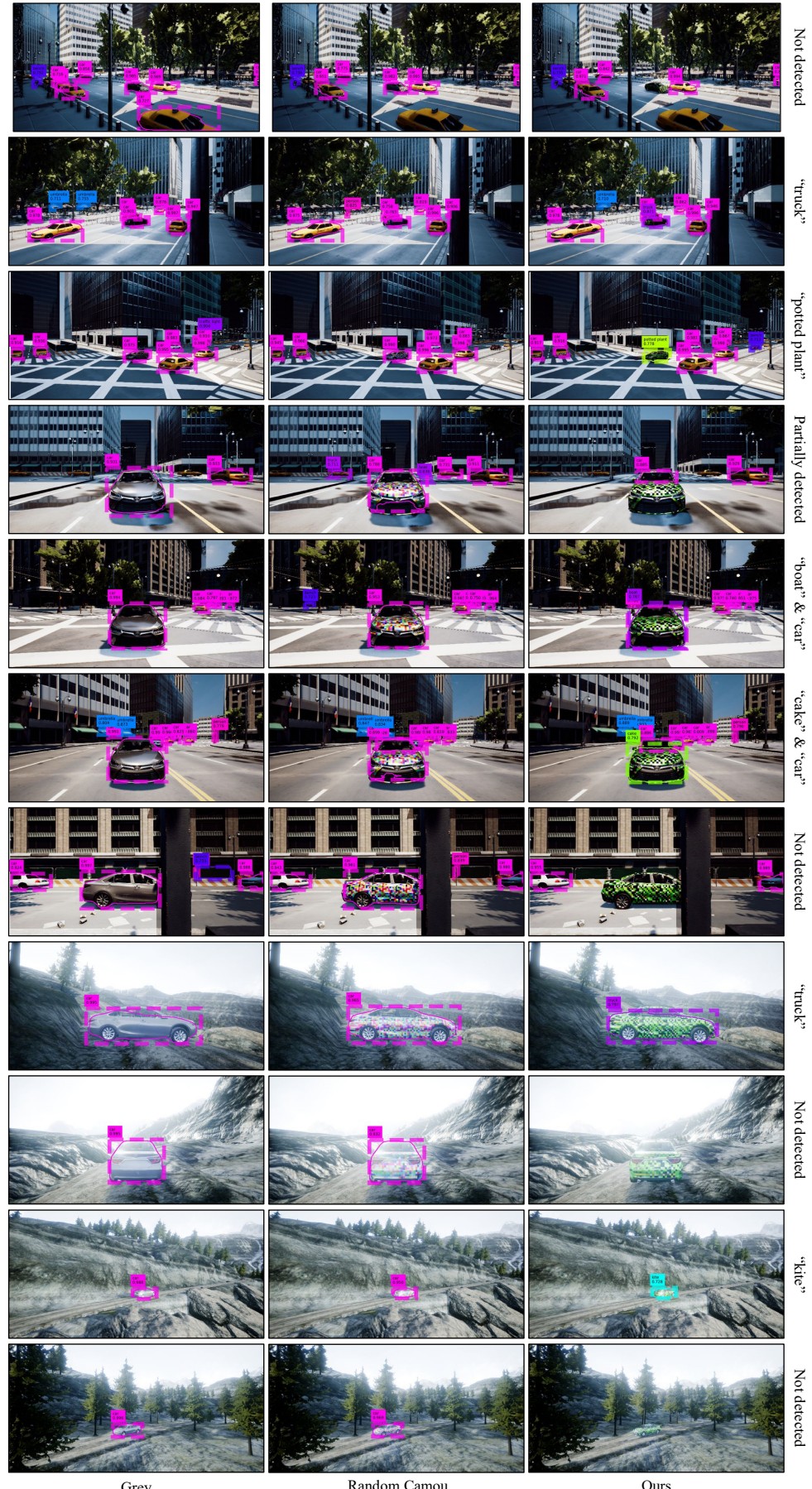

Figure 11: Qualitative comparison of the Mask R-CNN detections results of the grey baseline color, random camouflages and our learned camouflages in different transformations. Zoom in for more details.

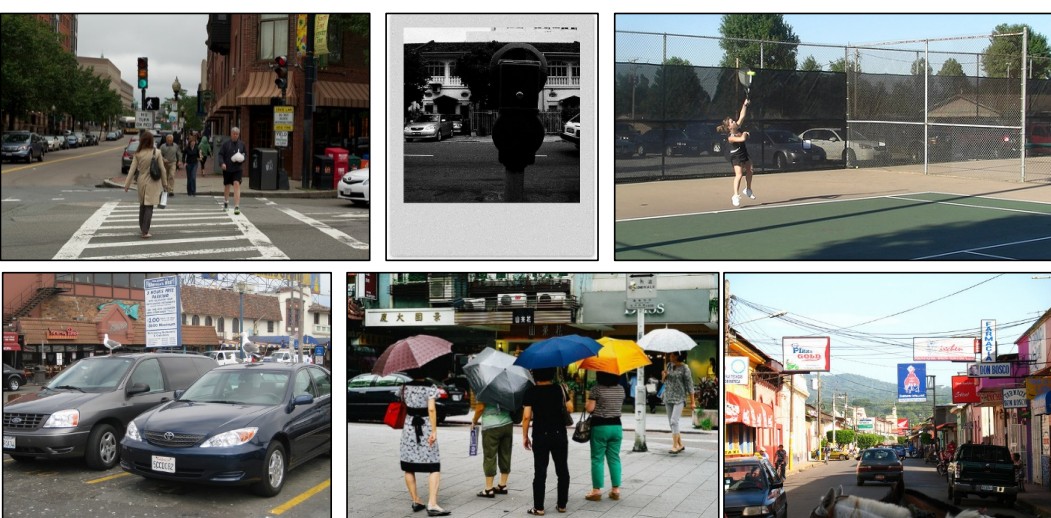

Figure 12: A fraction of different Toyota sedan appearances in MS COCO dataset.

