# OpenReview forum: "CAMOU: Learning Physical Vehicle Camouflages to Adversarially Attack Detectors in the Wild"
_ICLR.cc/2019/Conference_

### Official Review · AnonReviewer1 · 2018-10-29
**Adversarial attacks for vehicles in simulators**

**Rating:** 7
**Confidence:** 3

**Review:**

Adversarial attacks and defences are of growing popularity now a days. As AI starts to be present everywhere, more and more people can start to try to attack those systems. Critical systems such as security systems are the ones that can suffer more from those attacks. In this paper the case of vehicles that attack an object detection system by trying to not be detected are tackled.

The proposed system is trained and evaluated in a simulation environment. A set of possible camouflage patterns are proposed and the system learns how to setup those in the cars to reduce the performance of the detection system. Two methods are proposed. Those methods are based on Expectation over transformation method. This method requires the simulator to be differentiable which is not the case with Unity/Unreal environments. The methods proposed skip the need of the simulator to be differentiable by approximating it with a neural network.

The obtained results reduce the effectivity of the detection system. The methods are compared with two trivial baselines. Isn't there any other state of the art methods to compare with?

The paper is well written, the results are ok, the related work is comprehensive and the formulation is correct. The method is simply but effective. Some minor comments:
 - Is the simulator used CARLA? Or is a new one? Where are the 3D assets extracted from?
 - Two methods are proposed but I only find results for one

---

> ### Author Response · Authors · 2018-11-24
> **Response to Reviewer 1**
>
> Dear reviewer 1,
>
> Thank you for your positive comments. We will answer your questions and comments one by one.
>
> Q: Isn't there any other state of the art methods to compare with?
>
> A: Some of the existing works are closely related to ours but they are not directly applicable to our problem set, to the best of our knowledge. They are not applicable mainly due to two characteristics of our problem: 1) we aim to learn a single camouflage for the vehicle under a variety of transformations (location, camera view angle, lighting, etc.), and 2) many modules (painting the camouflage to the car body, photographing it, etc.) prevent the flow of gradients.
> Two blackbox attack methods [1] and [2] are especially relevant to ours. [1] trains a substitution network to mimic the behavior of the blackbox model to be attacked. However, their work requires the gradients to flow between the image and the perturbation. This requirement is not met in our problem because it is hard to use an analytic function to describe how the camouflage is converted to an image of the vehicle. [2] relies on accurate single time gradient estimation. However, randomness is inevitable (and is indeed implemented by the simulator) when we paint the camouflage to the car, drive it to different locations, take picture of the vehicle, etc.
>
> [1]Papernot, Nicolas, et al. "Practical black-box attacks against machine learning." Proceedings of the 2017 ACM on Asia Conference on Computer and Communications Security. ACM, 2017.
> [2]Chen, Pin-Yu, et al. "Zoo: Zeroth order optimization based black-box attacks to deep neural networks without training substitute models." Proceedings of the 10th ACM Workshop on Artificial Intelligence and Security. ACM, 2017.
>
>
> Q: Is the simulator used CARLA? Or is a new one? Where are the 3D assets extracted from?
>
> A: We build the simulated environments ourselves upon AirSim (https://github.com/Microsoft/AirSim). CARLA was one of our candidates when we started this project. However, we soon found that AirSim offers more flexible configurations and essential functionalities for our need. We took advantage of many of AirSim’s features and also implemented new features such as the external cameras, dynamic materials, texture communication via rpyc, etc. We purchased some 3D assets (e.g., Camry and the urban environment).
>
>
> Q: Two methods are proposed but I only find results for one.
>
> A: The two key techniques jointly make it possible to learn a single camouflage to physically fail the detectors. Please see Figure 4 for an illustration of the unified framework. Probably our text was not clear enough and caused the confusion; we will improve the clarity.

---

### Official Review · AnonReviewer3 · 2018-11-02
**Physical adversarial attack on object detectors is interesting.**

**Rating:** 8
**Confidence:** 4

**Review:**

This is an interesting paper targeting adversarial learning for interfering car detection. The approach is to learn camouflage patterns, which will be rendered as a texture on 3D car models in urban and mountain scenes, that minimizes car detection scores by Mask R-CNN and YOLOv3-SPP.

Different from image-based adversarial learning, this paper examines whether 3D car textures can degrade car detection quality of recent neural network object detectors. This aspect is important because the learned patterns can be used in the painting of real-world cars to avoid automatic car detection in a parking lot or on a highway.

The experimental results show that the car detection performance significantly drops by learned vehicle camouflages.

 Major comments:

- It is not clear how learned camouflage patters are different in two scenes. Ideally, we should find one single camouflage patter that can deceive the two or more object detection systems in any scenes.

Minor comments:

- In abstract, it is not good that you evaluated your study as "interesting". I recommend another word choice.

---

> ### Author Response · Authors · 2018-11-24
> **Response to Reviewer 3**
>
> Dear reviewer 3,
>
> We appreciate that you highlighted the significance of our work. For your question concerning the transferability of the camouflage, we did have one single camouflage tested in both environments. Probably our presentation was not clear enough and caused the confusion; we will improve the clarity. We transfer and test the single camouflage under different detectors, across different locations of an environment, and in two different environments. The learned camouflage outperforms the baselines despite some of the factors (detectors, locations, the mountain environment) are unseen during the training.
>
> More concretely, we learn a single camouflage on Camry in the urban environment against the Mask-RCNN detector. We then test this camouflage on Camry in the mountain environment against Mask-RCNN (Ours - Transferred in Table 4.), on Camry in the urban environment against the YOLO detector (Ours (Mask R-CNN trained) in Table 2.), on SUV in the urban environment against Mask-RCNN (Table 5.), and on Camry in the urban environment against Mask-RCNN with different cameras (Ours - Transferred in Table 6.). All of those results significantly outperform the baselines. It shows that it is possible to have a single camouflage generalize to different locations, detectors, environments, and even vehicles.

---

### Official Review · AnonReviewer2 · 2018-11-02
**Interesting problem, interesting approach but misses opportunities for detailed analysis. Not clear it will scale to real-world applications.**

**Rating:** 4
**Confidence:** 3

**Review:**

The authors investigate the problem of learning a camouflage pattern which, when applied to a simulated vehicle, will prevent an object detector from detecting it. The problem is frames as finding the camouflage pattern which minimises the expected decision scores from the detector over a distribution of possible views of this pattern applied to a vehicle (e.g. vantage point, background, etc). This expectation is approximated by sampling detector scores when applying the detector considered to images synthesised using a number of vantage points and scene contexts. In order to generate a gradient signal with respect to the camouflage applied (the simulated image rendering is non-differentiable) the approach considers learning a clone network which takes as input the camouflage pattern, the vehicle model and a given environment and outputs the vehicle detector’s devision values. The clone network and the optimal camouflage are learned alternately in order to obtain a relatively faithful approximation. The approach is evaluated in simulation using two standard object detectors (Mask R-CNN and YOLOv3) on two vehicle models over a range of transformations.

Pros:
———
- interesting challenge

- the clone network provides an interesting solution to the challenge of having a (non-differentiable) simulator in the loop.

Cons:
———
- the fact that this is done in simulation is understandable but to some degree negates the authors’ point that physicality matters because it is harder than images. How effective is a learned camouflages in reality? It would be great to get at least some evidence of this.

- if the sim-2-real gap is too big it is not clear how this approach could ever be feasibly employed in the real world. Some intuition here would add value.

- the approach presented critically hinges on the quality of the clone network. Some analysis of robustness of the approach with respect to network performance would add value.

- little is offered in terms of analysis of the camouflages generated. The three failure modes triggered and discussed are intuitive. But are there any insights available as to what aspect of the camouflages triggers these behaviours in the object detectors? This could also add significant value.

In summary, this work addresses an interesting problem but it is not clear how impactful the approach will be in real-world settings and hence how significant it it. Some of the more technically interesting aspects of the submission (e.g. the quality of the clone network, any learning derived from the camouflages generated) are not explored.

Misc comments:
- equs 1, 2 & 3: min_* should be argmin_*

---

> ### Author Response · Authors · 2018-11-24
> **Response to Reviewer 2**
>
> Dear reviewer 2,
>
> Thank you for your detailed reviews. We will address your concerns one by one.
>
> Q: the fact that this is done in the simulation is understandable but to some degree negates the authors’ point that physicality matters because it is harder than images. How effective is learned camouflage in reality? It would be great to get at least some evidence of this.
>
> A: Unfortunately, we do not have any concrete evidence yet. We did have the same concern and have studied two potential solutions: 3D print and hydrographics. We will try to include them in the future work.
>
>
> Q: if the sim-2-real gap is too big it is not clear how this approach could ever be feasibly employed in the real world. Some intuition here would add value.
>
> A: The simulation-to-real gap is bounded by the baseline results in some sense: painting the car bodies using some popular colors, the detection rate is about 75% (mIoU). Additionally, there are two immediate strategies, due to our empirical findings in this work, to improve the camouflage robustness for the purpose of real-world use.
> 1) More transformations. The original expectation-over-transformation [1] was proposed to solve the exact question: How do we generalize the simulated perturbation to the physical world? They show that providing more transformations leads to better simulation-to-real generalization. This is verified by our experiments of generalizing from the training locations to the previously unseen test locations, especially the transfer from the urban environment to the mountain environment. In the experiments of [1], the authors learn a perturbation applied onto the surface of a turtle in simulation against the classifier in different view angles. They find that, if sufficient view angles are presented, the perturbation also works on the surface of real-world 3D printed turtles against the classifier.
> 2) Direct physical optimization. What we proposed is a generic framework for learning a vehicle camouflage to fail detectors. This framework can be readily applied to the physical world as well. One can replace our simulated transformations with physical ones. For example, we can update the physical camouflage on a real car by either painting the car body or cover it by electrochromic materials [2]. After that, we may choose some representative locations and set up cameras, take pictures at different time window, annotate them, send them to the detector for evaluation, and subsequently minimize the loss. We note this is certainly feasible although it incurs significant financial costs.
>
> [1] Athalye, Anish, and Ilya Sutskever. "Synthesizing robust adversarial examples." arXiv preprint arXiv:1707.07397 (2017).
> [2] Mortimer, Roger J., David R. Rosseinsky, and Paul MS Monk, eds. Electrochromic materials and devices. John Wiley & Sons, 2015.
>
>
> Q: the approach presented critically hinges on the quality of the clone network. Some analysis of the robustness of the approach with respect to network performance would add value.
>
> A: Thanks for the suggestion. We have studied the camouflage quality vs. the number of simulation called in Appendix E in our paper. The plots show that as more new samples join, the clone network behaves more and more like the V_t() and hence generates better camouflages. This suggests that new camouflages help the clone network to calibrate and hence the alternative optimization is necessary.
>
>
> Q: The three failure modes triggered and discussed are intuitive. But are there any insights available as to what aspect of the camouflages triggers these behaviors in the object detectors? This could also add significant value.
>
> A: Thank you for this suggestion. We acknowledge that the neural network’s explainability and interpretability are an independent and challenging research topic [1]. State-of-the-art work [2] also focuses on training interpretable model instead of interpreting the existing models. The region-based networks are even harder to interpret given they yield structured predictions. We partially address this concern by visualizing the gradient heatmap of the best detector's classification score w.r.t. the input image. We found that the detector places its attention predominantly on the upper car body, i.e. roof, hood, windows, and trunk. This explains that our camouflage works better for the front view and rear view since trunk and hood camouflages are more visible. The camouflage is less effective for the side views where the detector places its attention on uncamouflaged windows. The visualization and detailed analysis have been added to the text.
>
> [1] Zhang, Quan-shi, and Song-Chun Zhu. "Visual interpretability for deep learning: a survey." Frontiers of Information Technology & Electronic Engineering 19.1 (2018): 27-39.
> [2] Zhang, Quanshi, Ying Nian Wu, and Song-Chun Zhu. "Interpretable convolutional neural networks." arXiv preprint arXiv:1710.00935 2.3 (2017): 5.
>
>
> Q: - eqs 1, 2 & 3: min_* should be argmin_*
>
> A: We have revised it.

---

### Author Response · Authors · 2018-11-24
**Paper revision: Two new experiments**

Dear all,

We have added two experiments to the appendix of the revised PDF. In the first experiment, we present the relationship between the newly added samples and the quality of the learned camouflage. We find that the more samples added, the learned camouflage is better since the clone network estimates the score better. In the second experiment, we present the detector's attention by visualizing the gradient heatmap of the detector's classification score w.r.t. The input image. We find that the detector mostly places its attention on the upper body of the car. This leads to better camouflage performance in the front and rear detector view since the upper body camouflages are more visible in these two views. We have also found a bug in our countryside environment evaluation. Our learned camouflages turned out to be more robust and transferable than we previously anticipated after fixing the bug. We have updated the results in Table.4. Finally, we have made some other minor revisions according to the reviewers’ suggestions.

---

### Public Comment · (anonymous) · 2018-12-21
**Paper misses recent prior work that is very closely related**

This work does not cite or compare with work that appeared almost 6 months ago on attacking object detectors in the physical world and showing transferability. The work of Eykholt et al show a similar camouflage attack in making a stop sign disappear. curious about the differences to this paper, and what intellectual contributions it provides over existing work.

Eykholt et al., Physical Adversarial Examples for Object Detectors, USENIX WOOT 2018.

https://www.usenix.org/conference/woot18/presentation/eykholt

---

> ### Author Response · Authors · 2018-12-23
> **Planar objects (e.g., the stop sign) vs. non-planar objects**
>
> Thanks for the pointer to [1]. We will add it to our final version. We did have cited a similar paper[2] which also aims to physically perturb the stop-sign detectors. We will be glad to discuss the differences between our work and [1,2]. The discussion will also facilitate the other readers to understand the motivation of our paper better.
>
> Good physical camouflages are supposed to fail object detectors for any images taken about the camouflaged object under all conditions: object-to-camera distance, background, lighting condition, view angle, etc. When we formalize the problem and try to optimize with respect to the camouflage, however, the “imaging function” which transforms the camouflage to camouflaged object and eventually various images is unknown. Hence, the key challenge to learning the physical camouflage is how to tackle this unknown imaging function.
>
> Both[1] and [2] perturb the detectors of stop signs which are planar objects whose images, under changes in camera geometry, are related by linear 2D projective transformations.  This is in contrast to non-planar objects (e.g., a car) whose images are related by more complex range-dependent nonlinear transformations. Hence, [1,2] are able to simplify such imaging function to projective transformations (cf. Section 4.2.2 in [1] and Section 4.1 in[2]) without breaking the gradient chain between the perturbation and the detector' output score. Complex nonlinear transformations, however, require a dedicated 3D simulation, for instance the one used in our paper. The 3D simulation breaks the gradient chain and turns the problem into a black-box optimization problem. In short, the non-planar objects break [1,2]'s premise since the approaches therein rely on functions that are differentiable with respect to the camouflage.
>
>
> [1] Song, Dawn, Kevin Eykholt, Ivan Evtimov, Earlence Fernandes, Bo Li, Amir Rahmati, Florian Tramer, Atul Prakash, and Tadayoshi Kohno. "Physical adversarial examples for object detectors." In 12th {USENIX} Workshop on Offensive Technologies ({WOOT} 18). 2018.
> [2] Chen, Shang-Tse, Cory Cornelius, Jason Martin, and Duen Horng Chau. "Robust Physical Adversarial Attack on Faster R-CNN Object Detector." arXiv preprint arXiv:1804.05810 (2018).

---

### Public Comment · (anonymous) · 2018-12-30
**Clarification on Algorithm 1, Line 7**

Very interesting work. I have a question regarding Algorithm 1 line 7. When you calculate the actual score by querying the original detection model on the newly found camouflage, what did you store as s'? Is it a list of the scores over various transformations? Is it the minimum of the score over various transformations? Is it the sum of the scores over various transformations? It seems to me that s' must be a scalar, since you're comparing it with s*. Can you please clarify this? Thank you!

---

> ### Author Response · Authors · 2018-12-31
> **s' is a list of scores**
>
> Thank you for your positive comments. s' is a list of scores over various transformations and s* is a scalar. Instead of comparing s' and s* directly, we are comparing the mean of s', notated as s'/|s'|, with s* in Algorithm 1 line 9. We will change s'/|s'| to mean(s') to avoid further confusion.

---

### Meta-Review · Area_Chair1 · 2018-12-14
**metareview: interesting approach**

**Confidence:** 4
**Recommendation:** Accept (Poster)

**Metareview:**

This work develops a method for learning camouflage patterns that could be painted onto a 3d object in order to reliably fool an image-based object detector.  Experiments are conducted in a simulated environment.

All reviewers agree that the problem and approach are interesting.  Reviewers 1 and 3 are highly positive, while Reviewer 2 believes that real-world experiments are necessary to substantiate the claims of the paper.  While such experiments would certainly enhance the impact of the work, I agree with Reviewers 1 and 3 that the current approach is sufficiently interesting and well-developed on its own.